# A Novel In Vivo Model for Multiplexed Analysis of Callosal Connections upon Cortical Damage

**DOI:** 10.3390/ijms23158224

**Published:** 2022-07-26

**Authors:** Ana González-Manteiga, Carmen Navarro-González, Valentina Evita Sebestyén, Jose Manuel Saborit-Torres, Daniela Talhada, María de la Iglesia Vayá, Karsten Ruscher, Pietro Fazzari

**Affiliations:** 1Laboratory of Cortical Circuits in Health and Disease, CIPF Centro de Investigación Príncipe Felipe, 46012 Valencia, Spain; agonzalez@cipf.es (A.G.-M.); cnavarrog@cipf.es (C.N.-G.); vevita@cipf.es (V.E.S.); 2Laboratory of Medical Imaging, CIPF Centro de Investigación Príncipe Felipe, 46012 Valencia, Spain; jmsaborit@cipf.es (J.M.S.-T.); miglesia@cipf.es (M.d.l.I.V.); 3Laboratory for Experimental Brain Research, Division of Neurosurgery, Department of Clinical Sciences, Lund University, BMC A13, 2184 Lund, Sweden; dtalhada@cnc.uc.pt (D.T.); karsten.ruscher@med.lu.se (K.R.); 4LUBIN Lab-Lunds Laboratorium för Neurokirurgisk Hjärnskadeforskning, Division of Neurosurgery, Department of Clinical Sciences, Lund University, 22184 Lund, Sweden

**Keywords:** brain damage, stroke, traumatic injury, neuronal regeneration, pre-clinical models, callosal neurons, cortico-cortical connections, perineuronal network, microglia

## Abstract

Brain damage is the major cause of permanent disability and it is particularly relevant in the elderly. While most studies focused on the immediate phase of neuronal loss upon injury, much less is known about the process of axonal regeneration after damage. The development of new refined preclinical models to investigate neuronal regeneration and the recovery of brain tissue upon injury is a major unmet challenge. Here, we present a novel experimental paradigm in mice that entails the (i) tracing of cortico-callosal connections, (ii) a mechanical lesion of the motor cortex, (iii) the stereological and histological analysis of the damaged tissue, and (iv) the functional characterization of motor deficits. By combining conventional microscopy with semi-automated 3D reconstruction, this approach allows the analysis of fine subcellular structures, such as axonal terminals, with the tridimensional overview of the connectivity and tissue integrity around the lesioned area. Since this 3D reconstruction is performed in serial sections, multiple labeling can be performed by combining diverse histological markers. We provide an example of how this methodology can be used to study cellular interactions. Namely, we show the correlation between active microglial cells and the perineuronal nets that envelop parvalbumin interneurons. In conclusion, this novel experimental paradigm will contribute to a better understanding of the molecular and cellular interactions underpinning the process of cortical regeneration upon brain damage.

## 1. Introduction

Brain injury (BI) is the major cause of permanent disability. The most common causes of BI are stroke, either ischemic or hemorrhagic, and traumatic injury, either closed or penetrating injury. Independently of the causes, the different brain injuries share some key pathological hallmarks [1,2,3,4,5,6] (Figure 1A). In the immediate or hyperacute phase of BI, excitotoxicity and the activation of cell death pathways lead to neuronal loss in the first hours after damage. The window for therapeutic intervention in this immediate phase is limited to a few hours. Next, in the acute phase, microglia, astrocytes, and peripheral immune cells unleash a complex cellular response aimed at removing damaged tissue and repairing the neural circuits [2,7]. Notably, excessive neuroinflammation may also contribute to neuronal damage, e.g., by increasing oxidative stress or the release of matrix metalloproteinases (MMPs) [8,9].

The following subacute phase is characterized by a reduction of the inflammation and by a phase of synaptic plasticity. This window of plasticity lasts approximately 1 month in mice and 3 months in humans. In this period, the neurons reactivate, at least in part, a developmental program to promote neuronal regeneration, axonal growth, and synapse formation. However, in the central nervous system (CNS), the intrinsic reactivation of developmental growth is only partial. In addition, extrinsic repulsive molecular cues in cellular debris and scar tissue further frustrate the process of neuronal regeneration. As a result, structural and functional recovery of the cortical neurons is very limited. Crucially, axonal regeneration and plasticity are mainly confined to the so-called perilesional area (PL) around the core of the cortical injury and it is causally linked to motor recovery [7,10,11]. The core of the lesion is characterized by neuronal loss and it is clearly demarcated in basic histological preparations. Conversely, the definition of the PL is less obvious and multiple approaches are found in the literature. Commonly, the PL is empirically defined as the area adjacent to the core [10,12,13,14,15]. Other studies use molecular markers, such as the inflammatory microglial marker Iba1, to further define the cellular environment of the PL [16]. Altogether, the dynamic interaction between multiple cell types and the complex stereology that defines the area of plasticity in the PL are still poorly understood.

Notably, multiple experimental approaches can be found in the literature to model and analyze BI, each with specific pros and cons. Some experimental models necessitate non-trivial surgical procedures such as the occlusion of cerebral arteries [3,17] or require specific machinery such as the fluid percussion injury device for traumatic injury [5]. Moreover, many studies focus on the immediate stage of neuronal death and utilize simple methods aimed at quantifying the volume of core lesion such as the TTC (2,3,5-triphenyltetrazolium chloride) staining. Here, we aimed at developing a workflow that could be as effective and consistent as possible to study this regenerative process. We reasoned that an optimal experimental paradigm to investigate brain regeneration should be simple, robust, and allow both tridimensional reconstruction and subcellular resolution. Therefore, we developed and characterized a model of BI that we refer to as Controlled Cortical Damage (CCD).

Our work builds upon previous models, such as the stab wound model generated by Professor Götz [18] and the cortical tracing technique developed by the lab of Professor Carmichael as starting points to develop our experimental workflow [16]. The CCD model is based on a simple mechanical lesion of the motor cortex, which requires minimal training and is robust and reproducible, allowing the use of a fairly limited number of mice. We combined this mechanical lesion with neuronal tracing. In short, our experimental paradigm for CCD includes: (i) tracing of cortico-cortical callosal projection neurons; (ii) a controlled mechanical lesion of the motor cortex; (iii) immunolabeling using multiple markers in serial sections; (iv) a semi-automated 3D reconstruction of serial immunolabeling; (v) characterization of the motor function after injury.

Notably, while in large injuries most PL is compromised [11], in our focal lesions we could precisely define different subregions in the area surrounding the core of the cortical damage using multiple molecular hallmarks. As a paradigmatic example, we investigated the remodeling of the perineuronal nets (PNNs) that surround a subpopulation of cortical interneurons in the PL. By multiple labeling in adjacent serial sections, we could define Iba+ and Iba1− PLs. Interestingly, we observed a sharp counter correlation between the labeling for active microglia (Iba1) and the marker for perineuronal nets Wisteria Floribunda Agglutinin (WFA). PNNs were almost unaffected in Iba1− PL. Conversely, PNNs were barely detectable in Iba1+ PLs.

In conclusion, our CCD model provides a robust and effective workflow to investigate plasticity and tissue repair after brain injury. This pipeline allows a 3D volumetric analysis and multiple labeling in serial sections to gain insights into the cellular interactions in the injured regions of the brain. We believe the CCD model provides a relevant contribution to investigating the complexity of BI and the interaction of different cell types in this process.

## 2. Results

### 2.1. Controlled Cortical Damage (CCD): Biological Paradigm and Methodology

Our aim was to develop a methodology that produces a localized and focal injury to study cortical regeneration and plasticity upon injury taking advantage of callosal connections as a model [16]. We referred to this experimental paradigm as CCD. The schema in Figure 1 outlines the milestones of this experimental paradigm.

### 2.2. Tracing of Cortico-Cortical Connections

First, we established the CCD model in young mice (3–4 months). To trace callosal projection, we performed stereotactic injection using AAV expressing GFP under the synapsin promoter [19] into the left (non-lesioned) primary motor cortex of the hindlimb (bregma 0.2; medio-lateral (ML) 1.5, dorso-ventral (DV) 0.5). Since at this anterior-posterior (AP) level the callosal connections run parallel to the coronal sections, we could observe in a single section the injection site (Figure 2A, upper panels) and the cortico-cortical projections through the corpus callosum to the contralateral target site (Figure 2A, lower panels). The labeling of GFP-expressing axons gave enough resolution to identify axons individually (Figure 2A, inset). Incidentally, using AAV as tracers gives the opportunity of performing gain and loss of function assays through the expression or depletion of specific genes of interest. This approach allows to study the soma and the axon of the same neurons separately in the same slices and could be useful to investigate the role of a given gene-of-interest these specific compartments.

In addition, we showed that the labeling could be combined with different tracers such as biotin dextran amine (BDA). Specifically, we labeled with BDA injection in the right motor cortex the ipsilateral connections from the right forelimb (bregma 1.60; ML 1.60; DV 0.5) to the right hindlimb. As for the AAV injection, this labeling provided a cellular resolution to study individual axons. Combining different tracers in different brain structures could be a useful tool for a better understanding of plasticity and connectivity in different neuronal projections in the same sample.

### 2.3. CCD in Young Mice

To induce the CCD, we performed a mechanical injury that targets the left hindlimb region. Briefly, using a motorized stereotactic apparatus, we performed a craniotomy and we introduced a metallic bar into the right primary motor cortex, crossing through the cortex (Figure 3A,B). This mechanical procedure was fast, simple, and with a minimal mortality rate (0 mice out of 11). One week after CCD, the brains were collected and processed in floating coronal sections, equidistantly distributed in the series. The injury volume was obtained by measuring the area of each affected slice [20]. In young mice, the average volume of the CCD was 1.26 ± 0.13 mm^3^ (Figure 3C). Overall, the procedure of the CCD model did not require complex training, was cost-effective, and provoked a focal and highly reproducible lesion compared to other models. Having obtained satisfactory results in young mice, we next tested the consistency of this model in aged animals.

### 2.4. Tracing Callosal Projections and CCD in Aged Mice

Brain damage is the leading cause of permanent disability. It is particularly relevant to the elderly [4,16,17,21]. Plasticity is reduced over time and restricted in old individuals, so it is crucial to understand histological changes upon brain damage in the aged population. To assess this objective, we characterized our CCD model in aged mice (9–12 months).

The schema of the experimental design is shown in Figure 4A. Briefly, 3 weeks prior to injury, we performed the tracing of callosal projections into the left primary motor cortex, injecting an AAV vector to express GFP. Then, we induced the mechanical injury as described above into the right primary motor cortex. To study functional motor outcome upon CCD, we performed a battery of behavioral tests 2 days before, 2 days after, and 7 days after the lesion. Finally, the brains were collected and processed in serial sections for further histological processing.

The quantification showed that the volume and the consistency of the CCD were similar to those obtained in young mice (Figure 4C, Appendix A). Again, the data show that the CCD lesion is highly reproducible compared to other BI models [2,5,21]. Notably, we did not observe mortality in this experimental batch in spite of the age of the mice (0 deaths out of 11 mice). Overall, the proposed methodology combining tracing and CCD provides an effective tool to investigate the cortical damage and the plasticity of cortico-cortical connections.

### 2.5. Motor Function Characterization upon CCD Model in Aged Mice

Another challenge in developing the CCD focal injury model was to provide measurable motor deficits in injured mice. The observation of the mice in standard housing conditions failed to reveal overt motor impairments upon CCD. Therefore, we performed a battery of behavioral tests to ascertain what tasks could provide a simple, sensitive, and reproducible functional readout. Considering injury size and localization, we predicted that the CCD would provoke a subtle motor dysfunction, mainly in the left hindlimb of affected mice.

First, we analyzed the behavior test 2 days before CCD, to define the basal state of mice. Next, we performed the following battery of tests 2 days (D2) and 7 days (D7) after CCD. In the treadmill test, mice are forced to run on a moving strip and filmed for a couple of minutes. This assay gives a general readout in terms of body symmetry and coordination. We found a slight unbalance at D2 in the injured group. This deficit was recovered at D7 (Figure 5A).

Next, we assessed motor dysfunction upon CCD in the ladder test. We used narrow- and wide-spaced horizontal ladders to further challenge the walking skills of the animals. In the narrow spaced ladder, the mice walk at a normal stride. The wide-spaced ladder forces the mice to stretch their stride to reach the next step of the ladder. The narrow-spaced ladder showed that injured mice performed worst at D2, but recovered at D7 to pre-injury levels (Figure 5B). Conversely, in the wide-spaced ladder, the mice with CCD showed obvious deficits both at D2 and at D7 (Figure 5C). Overall, both ladder tests were simple and reproducible. Increasing the spacing of the ladder was sufficient to improve the sensitivity of the test and reveal the deficits related to the focal injury of CCD.

Since the CCD model provokes an asymmetrical lesion on the right motor cortex, we tested the motor behavior of injured mice in the rotating pole test (RPT), an asymmetrical assay used in the sensorymotor characterization of stroke models [22]. Changing the direction and speed of the rotation provides a specific motor characterization in unilateral lesions. Here, we used a fixed speed of three rounds per minute (rpm), which was already challenging for aged mice. This assay was precise enough to show motor dysfunction both at D2 and at D7 upon CCD (Figure 5D). As expected, the mice performed better while the pole rotates to the right (counterclockwise), since the dysfunction of the affected left hindlimb (LH) is being compensated by the rotation. Altogether, the RPT was a particularly useful tool to study functional motor outcome in the CCD both in terms of sensitivity and simplicity.

Finally, we used the Catwalk XT technology, which is largely used to test motor phenotype in different animal models [23]. This technology offers an automatic gait analysis system, extracting multiple parameters related to paw placement or dynamics. For simplicity, here we provide data from four parameters in which we detected a relevant motor impairment induced by CCD (Figure 5E).

The results show that the max contact and print area of the LH decrease at D2 and D7 upon CCD induction. Furthermore, the swing and body speed decreased at D2. Nevertheless, the animals recovered from this impairment at D7 (Figure 5E).

In summary, the CCD focal injury provoked a subtle motor deficit that did not overtly affect basic motor functions such as body coordination or normal walking skills. Nonetheless, the CCD impaired the motor performance, particularly of the LH, in more challenging and sensitive experimental paradigms such as the wide-spaced ladder, RPT, and paw print area analysis.

### 2.6. Bioinformatic Pipeline of Tissue Processing and Stereological Analysis

To perform the stereological analysis, we developed a bioinformatics workflow to semi-automatically perform tissue reconstruction with a multiplex approach combining different immunostaining from the same mice in different series.

Briefly, the brains were cut in series and processed in floating sections. All the slices are equidistant between each other within a series, facilitating tridimensional reconstruction. Mounted sections were scanned using the Aperio Versa scanner to obtain a complete imaging of the slide. The images were opened in ImageJ to create the preliminary stacks (Figure 6A). To process the stacks, we designed a semi-automatic workflow that is divided into two different steps: sorting and alignment.

First, we developed an automatic algorithm in Python to sort the sections along the anteroposterior axis. In short, we manually thresholded the images in the DAPI channel. Next, the algorithm processes the images and calculates the area of the contour of the sections. Note that the contour step is important to compensate for the lack of tissue in the damaged area. Finally, the sections are automatically sorted in the stack assuming that the area of the contour will increase along the anterior-posterior axis (Figure 6B; Appendix A). Incidentally, this assumption is valid up to bregma −2.0. It follows that this algorithm worked nicely (~95% accuracy) in our region of interest that extends from bregma 2.5 mm to −1 mm, but it would not be effective for regions more caudal than bregma −2.0.

Our next challenge was the automatic registration/alignment of the sections. Different algorithms were previously developed for this aim such as the ImageJ plugin bUnwarpJ or AMaSiNe [24], which are based on elastic deformations of the images. In our experience, elastic image deformation worked very well in intact brain sections. However, the elastic transformation of injured brain sections often resulted in unsound artifacts that undermined the efficacy of these approaches for our purposes (not shown).

Therefore, we wrote an algorithm in IJ1 macro ImageJ code, which is based upon the plugin “Align image by line ROI” written by Johannes Schindelin (https://imagej.net/plugins/align-image-by-line-roi (accessed on 15 January 2020). This algorithm improves the usability of the original plugin and requires only minimal manual assistance to draw a reference line in each slice. Next, it takes advantage of a quadratic transformation to register the sections (Figure 6C; original code in Appendix A).

Overall, this semi-automatic workflow is an effective tool to correctly sort and align brain sections for stereological quantification. Once the series is registered, multiple series and immunolabeling can be combined for multiplexed analysis of the callosal connections and of the brain injury as we show in Figure 7 and Figure 8.

A typical experimental paradigm may entail cutting up to eight series of sections (e.g., 40 µm thick) from one brain. Since the DAPI channel is always used as a reference for the registration, we may in theory combine up to three different labelings per series in a standard microscope with green, red, and far red channels available. Notably, this approach allows reducing the number of animals utilized by combining different immunolabeling from the same brains.

### 2.7. Tissue Damage and Neuroinflammation upon CCD

As a first application of this workflow, we combined the volumetric quantification of the CCD injury with the analysis of the inflammation in the PL. Brain damage results in an inflammatory response that entails the activation of microglia and macrophages infiltrated due to blood barrier disruption. Iba1 is commonly used as a marker to label active microglia in the inflammatory environment. Microglia/macrophages contribute to debris clearance upon brain injury [5,8,25]. Furthermore, brain injury and the leakage of the blood –brain barrier induce the permeability of neuronal tissue that can be observed using IgG staining as a proxy [26,27,28]. Here, we combined IgG and Iba1 staining to evaluate tissue damage upon CCD.

As expected, the injured samples showed a greater intensity in both markers compared to the control group in the PL. To visualize the distribution of tissue damage and neuroinflammation in injured brains, IgG and Iba1+ areas were measured in each slice and represented in a Cartesian coordinate system (Figure 7C). The curve shows that tissue damage and microglia/macrophages population are delimited near the injured area upon CCD, without spreading along the anteroposterior axis. The volume of each staining was calculated by the integration of each polynomial fit equation of the areas calculated in the sections (Figure 7D).

Interestingly, the patterns of Iba1 and IgG positive regions were similar but not entirely overlapping (Figure 7A–C). For instance, we often found Iba1 labeling more intense close to areas where we could observe the presence of retraction bulbs from injured callosal neurons (labeled with yellow arrows in Figure 7B). We speculate that this may suggest that microglial cells may be involved in clearing cell debris upon axonal degeneration in these areas.

Incidentally, we did not detect major differences between young (3–4 months) and aged (9–13 months) animals in the volume of injury and of the Iba1 positive region (Appendix A). Nonetheless, we observed a tendency towards increased volume of the Iba1 positive region in aged mice (Appendix A). In future studies, it will be interesting to evaluate whether Iba1 activation in older mice (20–24 months) is increased. Altogether, this analysis indicates that CCD results in a focal and localized injury, causing tissue damage and inflammatory response fairly restricted to the core region.

### 2.8. PNNs Are Affected in the PL upon CCD

To further characterize the cellular alterations provoked by CCD in the perilesion, we investigated the remodeling of the PNNs. PNNs are a specialized form of extracellular matrix that specifically enwraps PV interneurons (Appendix A). On one hand, it has been previously shown that, upon CNS injury, PNNs are decreased in the areas around the lesion. This decrease appeared variable depending on the model and the methods used for quantification [12,13,14,15,29,30]. It is noteworthy that most of those studies simply define the PL as the area physically close to the injury and do not take advantage of other markers to define the PLs more precisely. On the other hand, macrophages were previously involved in the degradation of the extracellular matrix in a different cellular context such as tumor and wound healing [31,32].

To study the plasticity of PNNs upon CCD, we performed labeling with the PNNs marker with Wisteria Floribunda Agglutinin (WFA) staining in control and injured mice (Figure 8A). As expected, we observed a reduction of WFA labeling upon CCD in the PL. Interestingly, we noticed out of serendipity that this reduction in WFA staining mostly overlapped with regions with activated microglia in the series labeled for Iba1. Therefore, to quantify the WFA decrease in the PL, we combined information from the two series. We took advantage of the series labeled for Iba1 (Figure 7) to define in the PLs Iba1−positive (Iba1+) and Iba1−negative regions (Iba1−). Next, we quantified in the WFA-labeled series the number of WFA positive cells (WFA+) in Iba1+ and Iba1− PLs. Since WFA staining depends on the specific cortical region, we took as internal control the contralateral (non-injured) mirror regions of the cortex to quantify the decrease in the number of WFA+ cells in Iba1+ and Iba1− PLs (Figure 8C).

Overall, this analysis further supports the evidence that PNNs are remodeled upon injury in the PL. In addition, it suggests that Iba1+ cells may be involved in the degradation of PNNs.

## 3. Discussion

Since brain damage in humans is a very heterogeneous disorder, multiple mouse models were developed to investigate specific aspects of brain damage. Each model has specific strengths and weaknesses [3,5]. For instance, the middle cerebral artery occlusion (MCAO) model is a common model for ischemic stroke. This model is useful to mimic the lack of blood supply to the brain. However, it requires a non-trivial surgical operation and often affects big regions of the cortex together with the striatum and is associated with high mortality [2,17,21]. Other methods, such as stereotactic injections of collagenase or photothrombotic stroke, can be more easily targeted to specific regions of the cortex, but the outcome of these paradigms may be relatively variable in our experience. This variability increases the need for bigger experimental groups, which is expensive and time-consuming. Similarly, many models of traumatic injuries were developed to mimic various damages including blast waves, crush, or penetration by a projectile. These methods often require specific equipment and affect relatively big brain areas [5].

Our long-term goal was to investigate molecular and cellular mechanisms of tissue regeneration in the PL. Here, we aimed to induce a focal lesion in the motor cortex that could be as small as possible, while still provoking detectable motor deficits. We reasoned that a mechanical lesion guided by a stereotactic apparatus would be a simple, cost-effective, and reproducible approach to obtain this goal. Importantly, a more consistent model leads to a reduction in the number of mice used in line with the ethic of modern regulations on animal welfare. Therefore, we adapted to our aim the stab wound lesion protocol developed by Götz [18]. We referred to this experimental paradigm as CCD.

In the CCD model that we developed, we observed a minimal mortality rate (0 dead mice out of 22). The injury was fairly reproducible with regard to the size of the lesion, the histological hallmarks, and the behavioral outcome. In particular, the use of a stereotactic apparatus allowed us to target a very specific area of the brain. Incidentally, this approach may be easily adapted to specifically target other areas of the brain.

### 3.1. Behavioral Characterization

Another challenge that we faced in establishing the CCD model was the behavioral characterization of the motor deficits provoked by the CCD in the hindlimb motor cortex. In contrast to other models, the focal injury induced by the CCD causes a fairly minor impairment. In standard housing conditions, we did not observe an overt motor impairment. The battery of behavioral tests that we developed included multiple assays with different degrees of sensitivity. On one hand, we failed to observe major alterations in assays such as the treadmill test or the narrow-spaced ladder. This evidence suggests that the injury provoked by CCD is relatively minor. On the other hand, more sensitive assays such as the wide-spaced ladder, the Catwalk, and the rotating pole provided a convenient read out for the motor characterization of the CCD. Among the tests that we used in the current study, the wide-spaced ladder and the rotating pole tests were arguably the most effective.

Notably, in the rotating pole assay, the sensorimotor balance can be tested asymmetrically by inverting the rotation of the pole. Therefore, this motor test turned out to be very sensitive in evaluating the impairment induced by the unilateral cortical injury of the CCD. In addition, the equipment required for the rotating pole test is affordable and the analysis of the results was not very time-consuming. Altogether, this evidence suggests that the CCD fulfills our goal of obtaining a minimal lesion that provoked a motor impairment.

### 3.2. Pipeline for Image Processing: Volumetric and Histological Analysis

Another ambition of the methodology that we developed was to provide a flexible workflow that may allow a volumetric analysis of the injury while allowing cellular and subcellular resolution. To this aim, we combined the methodology of serial sectioning with an automated slide scanner and a semiautomated bioinformatic pipeline to sort and register the brain sections along the anterior-posterior axis of the bregma. Here, we performed a volumetric analysis of the core of the injury, of the area of inflammation using the marker for microglia/macrophages Iba1, and of the tissue permeability using IgG labeling as a proxy [27,28]. Concurrently, the same serial sections were also labeled for GFP to show the axons of the callosal neurons and for WFA, a marker for PNNs.

As proof of principle, our representative pictures showed that we could clearly visualize callosal axons and retraction bulbs around the PL in our histological preparations. In future studies, it will be interesting to exploit this methodology to investigate how callosal axons regenerate in the PL at different time points (e.g., D7 to D30) and if they are able to form putative synapses using multiple synaptic markers.

### 3.3. PNNs Remodeling and Inflammation in the PL

In addition, we investigated the remodeling of PNNs in the PL upon CCD. PNNs are a specialized extracellular matrix that envelop specific subtypes of cortical interneurons. While the function of PNNs is not entirely understood, different studies showed the formation of PNNs in cortical interneuron counter correlates with neuronal plasticity during development [33,34]. Interestingly, enzymatic degradation of PNNs reactivates plasticity in mature neurons [33]. Moreover, previous studies suggested that the PNNs decrease in the PL upon brain damage. These studies used different methodologies and observe a decrease in the number of PNN-positive cells that vary from 25% up to 60% of the control [12,13,14,15,29,30]. Notably, in most of the studies that we could find in the literature, the PL is empirically defined as the area in proximity to the core of the lesion. Here, we analyzed the plasticity of PNNs using WFA labeling in different serial sections. The same series used for the volumetric analysis of the inflammation were used in this analysis to define the Iba1−positive and -negative regions in the PL. Interestingly, we found that WFA staining was strongly reduced in Iba1+ PL (~80% reductions as compared to the control). In some samples, the WFA labeling was barely detectable in Iba1+ areas. Conversely, WFA labeling was barely affected in Iba1− PL areas. We speculate that this counter correlation between Iba1 and WFA may suggest that activated microglia could directly contribute to the degradation of the PNNs. In fact, it was shown in other cellular contexts that innate immune cells such as monocytes and macrophages [35,36] produce enzymes that degrade the extracellular matrix including MMP9.

## 4. Materials and Methods

### 4.1. Animals

In this study, we used young (4 month-old) and aged (9–12 month-old) C57BL/6 female mice. The animals were kept and cared in the animal facility of the CIPF Centro de Investigación Príncipe Felipe (Valencia, Spain). Experiments were supervised by the bioethics committee of the institute and performed in compliance with bioethical regulations of the European Union and Spain. Animals were group-housed with food and water ad libitium in standard housing conditions.

### 4.2. Experimental In Vivo Model of CCD

To assess fair bioethical practices, analgesia was provided to mice, administrating 100 µL of morphine (1.5 mg/mL) 30 min before the surgery and buprenorphine (0.03 mg/mL) 6 h later and the next morning. Moreover, weight loss was monitored along the experiment as an early humane endpoint parameter. Thus, animals that presented a body weight loss above 20% would be sacrificed and excluded from the study, which was not the case.

Stereotactic injection of adenoassociated-viral vector for the expression of GFP was performed to trace axonal projection of the corpus callosum as previously described [19]. Briefly, mice were placed in a stereotaxic frame, under isofluorane anesthesia with the skull exposed. To target the primary motor cortex, the selected coordinates (mm) relative to the bregma were as follows: anteroposterior, 0.2; lateromedial, 1.5; and dorsoventral, −0.5. Then, the Hamilton syringe containing the virus was introduced 1 mm into the cortex to produce a pocket. Two minutes later, the syringe was risen up 0.5 mm and 1 µL of the virus (diluted 1:4) was injected at a flow rate of 0.1 µL/min. After the injection, the needle was left up to 3 min for an appropriate virus diffusion.

For BDA injection (Dextran biotin 3000 MW, Fisher Scientific, Hampton, NH, USA), the stereotactic injection was as previously described with some modifications. Briefly, we diluted 100 ug/µL in PBS and injected 1 µL at a flow rate of 0.1 µL/min. In this case, we trace ipsilateral neurons of the prefrontal cortex to observe other brain areas projections. The coordinates (mm) relative to bregma were as follows: anteroposterior, 1.6; lateromedial, −1.6; and dorsoventral, −0.5.

To perform the controlled cortical damage (CCD), a 2.1 mm diameter drill tip was introduced in the right primary motor cortex at a constant speed of 0.5 mm/s, using a motorized stereotaxic instrument to reduce injury variation (Stoelting, 51730M). Considering the size of the drill, the coordinates relative to bregma (mm) suffer a slight anteroposterior modification, compared to the ones used for AAV injection: anteroposterior, 0; lateromedial, −1.5; and dorsoventral, −2.5. The drill was introduced into the brain 3 times, waiting 2 min with the drill inside after the first immersion.

### 4.3. Motor Behavior Analysis

Motor functional outcome was tested using a battery of different behavioral motor test, described below. Considering the increased difficulty in the rotating pole and ladder, mice were trained 3 consecutive days before recording. To ensure that the motor deficits were linked to cortical damage, mice performed the test a couple of days before the CCD surgery. Since the model proposed involved a unilateral lesion in the right motor cortex, we previously checked whether animals showed side preference before the lesion induction. We evaluated the left and right hind limb data obtained in the rotating pole test (RPT) before the injury. The comparison between both paws was tested in rotation to the right and to the left modes (two-way ANOVA with Šídák’s multiple comparisons test, adjusted *p* = 0.809 and adjusted *p* = 0.644, respectively). The data did not reveal statistical differences, so we discarded hind limb dominance effect in our experimental groups. All the tests were done 2 and 7 days after the surgery to observe an asymmetrical motor deficit due to a unilateral CCD model. Except for the Catwalk XT data, the results were analyzed by 2 different researchers in a blinded fashion.

Rotating pole test:

The rotating pole test is a widely used test to assess motor dysfunction in unilateral cortical injuries, providing an effective and sensitive way to detect neurological motor deficits after CCD. The protocol used was previously described [22]. Briefly, mice walked through a wooden pole elevated 1 m height from the table. The pole can display 3 different modes: 1 static mode, in which it remains stopped, and 2 rotation modes to the right or left direction at 3 rpm speed. It is important to maintain always the same order of performance, from the easier mode (static) to the most challenging one (rotation to the left, considering that the CCD is done on the right primary motor cortex, so the left hindlimb will be the most affected).

Mice were recorded during the task performance to assess functional motor outcome, using a 0 to 6 scoring method, based on the walking ability and the number of slips per paw, as previously described [22] (see Appendix A).

### 4.4. Catwalk XT

The Catwalk XT system (Version 10.6, Noldus, Wageningen, The Netherlands) consists of an enclosed walkway on a glass surface, illuminated with specific light combination. Hence, once mice walk through the structure, each footprint is perfectly detected. After the recording, the software assesses a semi-automatic analysis to study a huge range of parameters, including paw position, walking pattern, and other run statistics (Catwalk XT 10.6 Reference Manual).

The settings were previously customized for mice, as well as the number of animals and time-point groups. The experimental and detection settings used to perform the experiments are shown in Appendix A. Three compliant runs were automatically analyzed by the software, using the “auto-classify” option. Note that it is important to stablish a proper intensity threshold that should be maintained for all the runs in the experiment. The result statistics are automatically assessed by the software, offering the mean and sem in the desired selected parameters. In this case, we focused on those which offer a greater phenotype: max contact area (mm^2^), print area (mm^2^), swing speed (cm/s), and body speed (cm/s). All the data presented refer to the left hind limb (LH), which should be the most affected according to the CCD coordinates.

Max contact area (mm^2^): refers to the maximum area of a paw touching the glass.

Print area (mm^2^): the surface of the complete paw print.

Swing speed (cm/s): the velocity of the paw during the swing, which refers to the part of the step cycle in which the paw is not touching the glass.

Body speed (cm/s): the velocity of a paw during the step cycle, which refers to the time travelled since the first contact with the glass until the next contact with the same paw.

### 4.5. Narrow and Wide-Spaced Ladder

Another way of analyzing motor impairment is skilled ladder walking for detection of (1) paw symmetry and (2) number of mistakes per paw. To assess this aim, we take advantage of the MotoRater structure, which consists of an enclosed transparent walkway with two mirrors in the upper part to observe the animal from 3 sides simultaneously. In the bottom part, a high-quality camera is installed to record the runs (MotoRater TSE systems). Briefly, mice walked 3 times over two ladders, which differed in bar spaced distance (1 or 3 cm, respectively) to increase the task difficulty, offering a higher sensitivity to motor deficits detection. The videos were analyzed using a 0 to 5 scoring system, considering number of mistakes and walking symmetry. This last parameter is based on the fact that mice place the hind limb in the same position where the front limb was previously placed.

### 4.6. Treadmill

Using the treadmill test, mice are motivated to run to avoid touching the electrified grid located at the back of the lane. Briefly, mice were placed in the lane for a quick training of 30 s at 15 cm/s (not evaluated). Later, the lane speed is changed to 20 cm/s for 1 min and to 25 cm/s for two minutes. These two different slow and fast modes are evaluated on the recorded videos, producing a highly challenging task to the animal. In contrast to the analysis of the previous test, in this case just 1 run per mouse was analyzed.

### 4.7. Immunohistochemistry

Eight days after CCD, mouse brains were processed as previously described (Navarro-González, et al., 2019). Briefly, mice were transcardially perfused with 4% paraformaldehyde to collect the brain, following a 2 h postfixation and cryoprotected in 30% sucrose for 72 h. The brains were sectioned with a cryostat at 40 µm, distributing the slices in 8 different anteroposterior ordered series for 3D reconstruction analysis.

Primary and secondary antibodies used were diluted in PBS with 0.25% Triton and 4% BSA. Incubation with the primary lasted for 48 h, while the secondary antibodies were left overnight at 4 °C in agitation conditions. The antibodies and dilution used were as follows: anti-GFP (1:1000, GFP-1010, Aves Lab), anti-Iba1 (1:500, 234004, Synaptic systems), anti-parvalbumin (1:250, 195002, Synaptic systems), anti-WFA biotinylated (1:1000, L1516-2MG, Sigma Aldrich, St. Louis, MI, USA); anti-chicken 488 (A-11039, Thermofisher, Waltham, MA, USA), anti-mouse 555 (for IgG quantification, A31570, Invitrogen, Waltham, MA, USA), anti-guinea pig 647 (A21450, Invitrogen), anti-rabbit 555 (A31572, Life Technologies, Carlsbad, CA, USA), and CyTM5 streptavidin (PA45001, GE Healthcare, Chicago, IL, USA). All the secondary antibodies were diluted 1:500. Fluorescence imaging of the whole slides was performed using a Leica Aperio Versa at 10× Plan Apo, selecting each slice on the slide separately.

### 4.8. Bioinformatic Image Processing

To perform 3D reconstruction and multilabeling approaches, an important step of imaging analysis is the image processing. Thus, we used Anaconda3 to sort the samples following an anteroposterior order and Fiji software to create and align the image stack.

Briefly, the pictures obtained in the Leica Aperio Versa are organized into a stack, separating each fluorescence channel into a specific one to facilitate the following workflow. To sort the images, firstly it is necessary to create a binary mask in Fiji where the complete slice area is selected on the DAPI stack. Next, a Gaussian blur filter (sigma = 3) is selected to homogenize the slice surface. Later, the mask is fixed manually, establishing a threshold that covers all the slice area (Figure 6B). The mask is applied to the whole stack and saved on the same folder with the remaining channel stacks. It is important to note that, for code sake, all the stack should be organized in the same folder per brain and named as follows: Experiment name_**Brain number**_ **Channel name**_mask (if necessary), highlighting in bold words the information required for a correct code running.

The theoretical statement of the code is based on sorting the sample considering that along the anteroposterior axes the slice areas increase following a logarithm curve. Hence, the code calculates the area of the DAPI mask by profiling and contouring the slice borders (shown in Figure 6B). Considering that the CCD generally produces a hole on the tissue, it is crucial that the contour processing goes above the injury so the sorting is not affected by the absence of tissue. Once the area is calculated, the code automatically transfers this information to sort the rest of the channels. It is worthy to mention that in our experiments the selected region of interest expands from 2.5 mm to −1 mm relative to bregma coordinates (see Allen brain atlas; https://mouse.brain-map.org/static/atlas (accessed on 15 January 2021), where this theoretical principle applies. If the area of interest belongs to posterior samples, then the code should be optimized. The rate of sorting efficiency with this tool is 95% accurate. Nevertheless, additional checking is suggested in case of broken slices or other artefacts related to tissue manipulation that could change slice dimensions. For further details in the code script and requirements, see below in Appendix A.

Lastly, the samples are finally aligned, using an adaptation of a previously published Fiji plugin named “Align image by line”. This tool allows the alignment of all the slices from the same stack, after providing the midline of each sample and selecting the line of reference to align the rest. This step is also automatized using a macro, which is shown in Appendix A. As a previous requirement, the macro should be customized considering the number of slices that contain each brain stack.

### 4.9. Stereological Measurements and Image Analysis

From sorted and aligned stacks, the quantification of injury volume, inflammation, and tissue damage impact is done. The methodology is described in [20]. Briefly, the volume of the injury was lately calculated by the integration of its polynomial fit equation, according to the formula: V = $\int_{a}^{b}A\left(x\right)dx$ in which A is the cross-sectional area and *x* represents the width of the interval [a,b].

To estimate inflammation and tissue damage, Iba1 and IgG staining pictures were used, respectively. Briefly, Gaussian blur filter (sigma = 5) was set to facilitate the homogenization and detection of Iba1 and IgG area. After that, a threshold is manually set, selecting the value that better fit in the whole samples, according to the contralateral hemisphere. The image is converted into a binary mask in which the region of interest per slices is automatically identified using the function named “Analyze particles” (size = 1000-infinity). The area was calculated on the region of interest detected by this tool. From these values, the volume of each parameter was calculated by integration of the polynomial fit equation, as previously described.

To analyze the cell-to-cell interaction between microglia and PNN upon CCD, the number and intensity of WFA+ cells was analyzed in the PL, determined by Iba1+ labelling. Briefly, rectangles in both Iba1+ and Iba1− (somatosensory cortex) were selected as regions of interest (ROI) in the ipsilateral and contralateral cortex. To delimit the number of WFA+ cells, the same threshold was established on both ROIs in the Iba1+ or Iba1− area, according to contralateral staining which conforms the intrinsic control in each sample. Once the threshold is selected, we automatically detected the contour of the WFA+ cells, using the tool “Analyze particles” (size = 200-infinity) of ImageJ. The ROIs of each cell were saved and the intensity was measured in the raw images. The data is normalized to the respective contralateral ROIs for each sample, showing the fold change in the graph (mean ± sem).

### 4.10. Statistical Analysis

Statistical analysis and graph preparations of all the figures were done using Graph Pad Prism 9 software. The data is represented as mean ± SEM. Significance is indicated with an asterisk in each figure legend, considering *p* ≤ 0.05 as significant. The statistical test used for each analysis is mentioned in each graph. Briefly, a two-way ANOVA with repeated measurements was used for narrow and wide-spaced ladder and treadmill to compare both control (Ctrl) and injured (CCD) groups along the different time-points (pre-injury: pre; D2 after CCD: D2; and D7 after CCD: D7). For the RPT data, we used a 2-way ANOVA with repeated measurements to compare how the CCD group performed the task in the two different models (rotation to the right and rotation to the left) along the different time-points. For the four parameters obtained through the Catwalk XT, a one-way ANOVA was used. Finally, to analyze the perineuronal nets upon CCD, we used a paired t-test to compare the relative fold change of cell number decrease, related to each contralateral area. The slices were considered as the N, according to the variability of the WFA expression pattern along the brain.

## 5. Conclusions

In conclusion, we describe here a novel experimental paradigm to investigate brain injury. This model that we refer to as CCD is based on a focal mechanical lesion of the cortex. The CCD model allows a stereotactic precision of the site of injury that can be combined with the tracing of the contralateral callosal projections. In addition, we developed a flexible and cost-effective pipeline that allows a volumetric analysis of the injury together with a cellular analysis of the cortical tissue in multiple series. In addition, we performed an extensive behavioral characterization of the motor deficits induced by CCD. Finally, we showed how PNNs were affected in the PL around the CCD specifically in Iba1+ regions. In future studies, we believe that this model will be particularly useful to investigate the plasticity of cortico-cortical callosal connection upon damage.

## 6. Future Perspectives

In our opinion, the field of neuronal regeneration and plasticity upon brain damage will grow and will be very impactful in the near future. On one side, the societal impact of brain damage, the major cause of permanent disability, is huge and will further increase with the aging of the population. On the other side, the last decades showed remarkable progress in the molecular understanding of the molecular biology of axonal growth and synapse formation during development. Thus, it will be interesting in the future to test the working hypothesis that some of those developmental pathways are at least in part involved in the process of neuronal plasticity upon damage. The development of specific pre-clinical models, such as the CCD model presented here, will be instrumental to achieve this aim.

While an extensive discussion of the main future challenges for the field would require a review on its own, we mention here a few key topics that we consider particularly relevant to acquiring robust knowledge of neuronal regeneration in the brain and in the cortical circuits.

In addition to axonal growth and synapse formation, the physiology of cortical circuits is further regulated by the process of axonal myelination. Deficits in the myelination were related to impaired signal transmission and were implicated in both neurodevelopmental pathologies and brain damage [37]. The process of re-myelination during the recovery and plasticity of cortical circuits is poorly understood and the oligodendrocytes are a major cellular target for novel therapeutic approaches to brain injury.

Another major challenge will be understanding the cellular and molecular mechanisms related to neuronal plasticity upon injury in the old. Multiple pathways related to survival, oxidation, and inflammation are altered in neurons by aging. However, these studies require the use of mice at least 18–24 months of age, which significantly complicates the experimental procedures and planning. Nonetheless, the development of novel preclinical models of brain injury and recovery in aged mice will be essential, considering the epidemiological impact of brain damage and related permanent disabilities in the elderly.

## Figures and Tables

**Figure 1 ijms-23-08224-f001:**
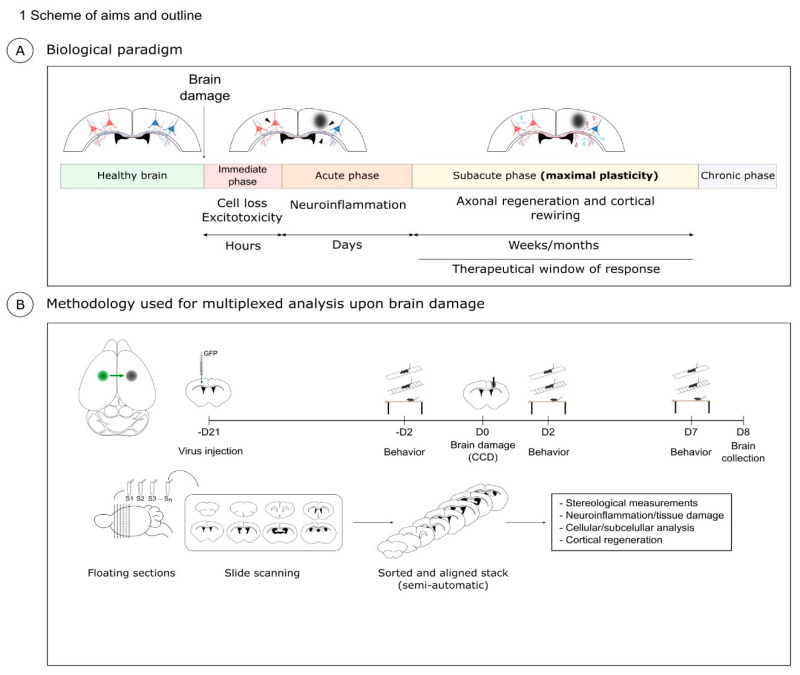
Controlled cortical damage (CCD): biological paradigm and methodology. (**A**) The cartoon represents the main phases that characterize brain injury: immediate (first hours), acute (up to 7 days), subacute (weeks–months), and chronic phase. During the subacute phase, maximal plasticity window takes place, representing a strategic therapeutic target to ameliorate disabilities linked to BI. (**B**) The scheme resumes the different steps of the CCD model. Firstly, the virus injection is performed to trace callosal projection in the hindlimb motor cortex (coordinates used relative to bregma: AP 0.2, LM 1.5, DV −0.5). Three weeks later, we induced brain damage in the right primary motor cortex affecting the left hind limb (coordinates used relative to bregma: AP 0, LM −1.5, DV −0.5). The motor phenotype is analyzed pre-, 2, and 7 days after the injury. Finally, brain samples are collected and processed to perform immuno-labeling of different markers to carry out stereological analysis for tissue regeneration.

**Figure 2 ijms-23-08224-f002:**
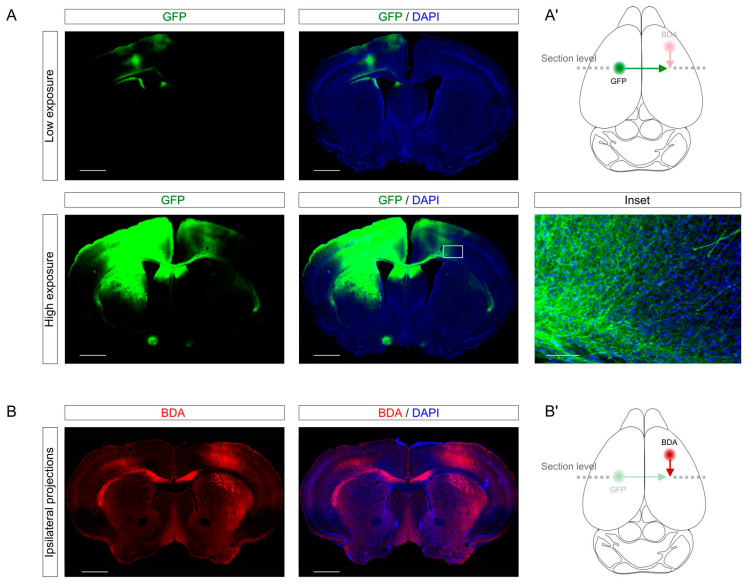
Tracing of cortico-cortical connections. (**A**) The images show the tracing of cortico-cortical projections from the left to right hindlimb motor cortex (bregma: AP 0.2, LM 1.5, DV −0.5). The upper row shows low-exposure images to see the virus injection site. The lower row shows high exposure to see the axonal sprouting in the contralateral cortex. The boxed area is magnified in the inset. (**A’**) The cartoon illustrates the schematic location of the contralateral callosal projections (green). In transparency (pink), the ipsilateral connections are shown in (**B**). The dashed line indicates the rostrocaudal level of the section. (**B**) We show here a different tracing technique using BDA. The coordinates of injection were in the prefrontal ipsilateral cortex to target axonal sprouting from the right forelimb to the right hindlimb motor cortex (bregma: AP 1.6, LM 1.6, DV −0.5). (**B’**) The cartoon shows the location of the ipsilateral cortico-cortical connections (red). In transparency (green), the contralateral connections are shown in (**A**). The dashed line indicates the rostrocaudal level of the section. Scale bar in (**A**,**B**), 1 mm; Scale bar in the Inset in (**A**), 100 µm.

**Figure 3 ijms-23-08224-f003:**
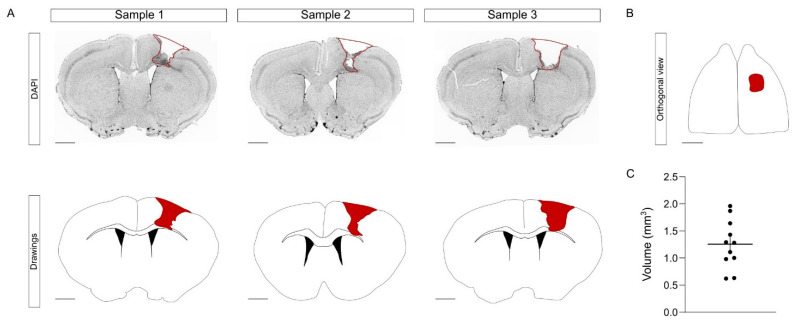
CCD in young mice. (**A**) In the upper row, representative images of the CCD injury in three different animals. In the lower row, the drawings represent the same slices presented above. The area of the lesion is highlighted as a red line above and as a red area below. Scale bar, 1 mm. (**B**) The serial organization of the slices allows to provide an orthogonal view of the injury. Scale bar, 1 mm. (**C**) Graph shows injury volume quantification in young mice. The graph shows mean ± sem; *n* = 11 mice.

**Figure 4 ijms-23-08224-f004:**
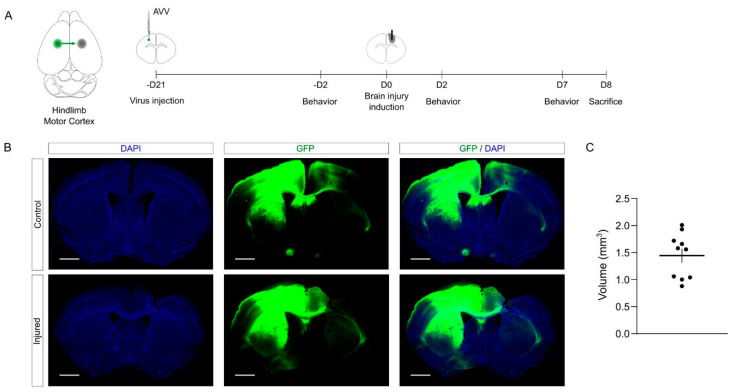
Tracing callosal projections and CCD in aged mice. (**A**) On the right, the cartoon shows the GFP tracing in shown green while the location of the injury is in gray. The schema shows the pipeline of the experimental approach followed in the characterization of CCD model in aged animals. (**B**) Representative images of control and injured brains, combined with the GFP tracing. Scale bar, 1 mm. (**C**) The graph shows injury volume quantification in aged mice. Mean ± sem. *N* = 10.

**Figure 5 ijms-23-08224-f005:**
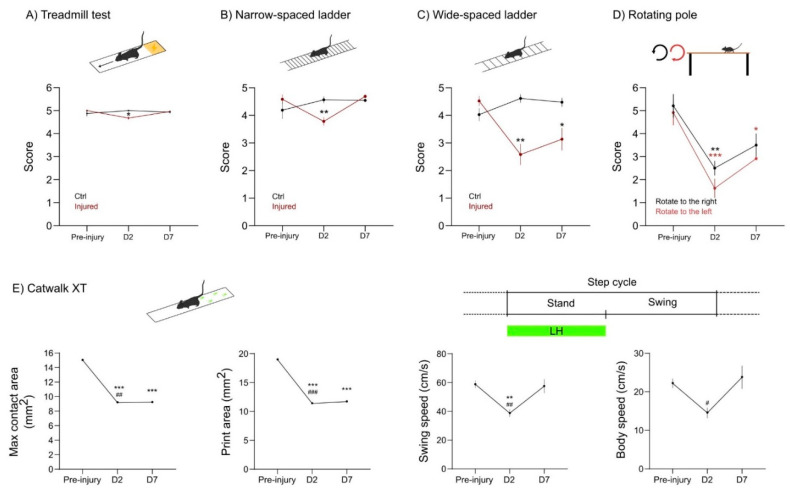
Motor characterization of CCD model in aged mice. (**A**–**C**) The graphs shows the score of control and injured animals in the multiple assays. The motor tests were performed at three different time-points: pre-injury (D-2), at 2 days (D2), and at 7 days (7D) after injury. (**A**) Treadmill test. (**B**) Narrow-spaced (**C**) and wide-spaced horizontal ladder. Control, *n* = 4; injured, *n* = 7 mice. Average ± sem. * *p* ≤ 0.05, ** *p* ≤ 0.01, two-way ANOVA with repeated measurements and Sídák’s multiple comparison test. (**D**) The graph shows the score of control and CCD mice in the rotating pole test (RPT), using two different rotation modes: rotation to the right (counter clockwise, black line) and to the left (clockwise, red line). *n* = 4 mice. Average ± sem. * *p* ≤ 0.05, ** *p* ≤ 0.01, *** *p* ≤ 0.001, relative to pre-injury. Two-way ANOVA with repeated measurements and Dunnett’s multiple comparison test. (**E**) The graphs show the quantification of the parameters indicated in the labels in the Catwalk XT assay pre-injury (D-2), at 2 days (D2), and at 7 days (7D) after injury. Average ± sem. ** *p* ≤ 0.01, *** *p* ≤ 0.001, relative to pre-injury; # *p* ≤ 0.05, ## *p* ≤ 0.01, ### *p* ≤ 0.001, relative to D7. One-way ANOVA and Holm-Šídák’s multiple comparisons test.

**Figure 6 ijms-23-08224-f006:**
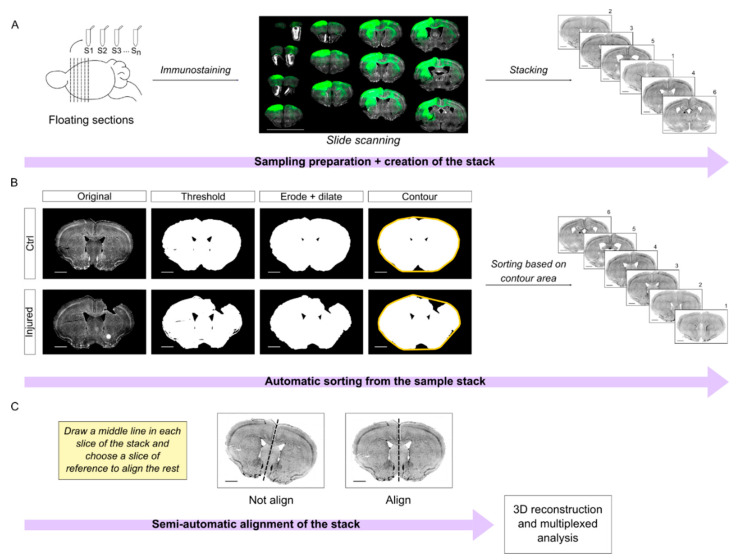
Bioinformatic pipeline of tissue processing and stereological analysis. (**A**) Steps of the workflow from tissue processing to image stacking. Briefly, the brains are processed in serial floating sections. The pictures of the immuno-labeled slides are taken with an automated slide scanner. Using ImageJ, the images of separated slices are grouped into a stack. Scale bar of full slide scanning, 10 mm; scale bar of individual slices, 1 mm. (**B**) Sample sorting and registration procedure using a code developed in Python. From DAPI staining images, a threshold is manually established to cover all the slice area. Then, the algorithm smooths tissue edges and calculates the contour area of each slice (yellow line). The area value determines the sorting along the anteroposterior axis for each section. Scale bar, 1 mm. (**C**) Manually tracing a reference line in all the slices of the stack and automatic alignment of the stack by ImageJ macro. Scale bar, 1 mm.

**Figure 7 ijms-23-08224-f007:**
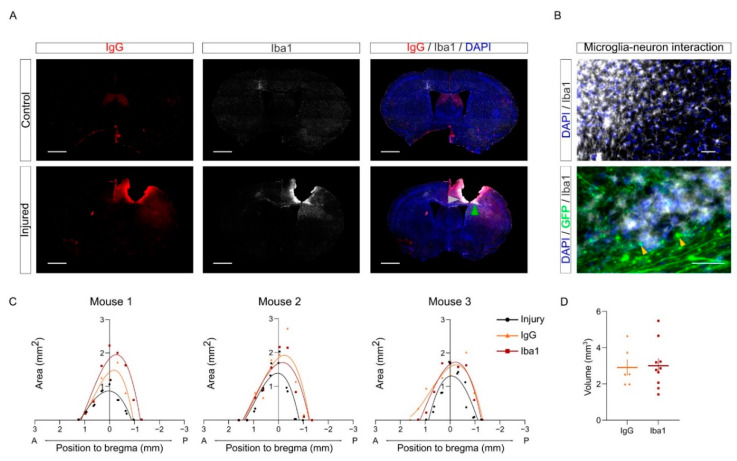
Tissue damage and neuroinflammation upon CCD. (**A**) Representative images of control and injured animals labeled for IgG and Iba1 markers upon CCD. Gray arrowhead, upper panel in (**B**). Green arrowhead, lower panel in (**B**). Scale bar, 1 mm. (**B**) Magnified areas from an injured sample are represented to show Iba1 and GFP staining in detail (indicated in merged slice with grey and green arrows). Scale bar, 50 µm. (**C**) Cartesian graphs represent data of injury (black), IgG (orange), and Iba1 (red) staining areas of three different animals, positioned relative to bregma coordinates. (**D**) Graph shows the volume quantification of IgG and Iba1 labeling in the injured mice. IgG, *n* = 6 animals; Iba1, *n* = 10 animals. Average ± sem.

**Figure 8 ijms-23-08224-f008:**
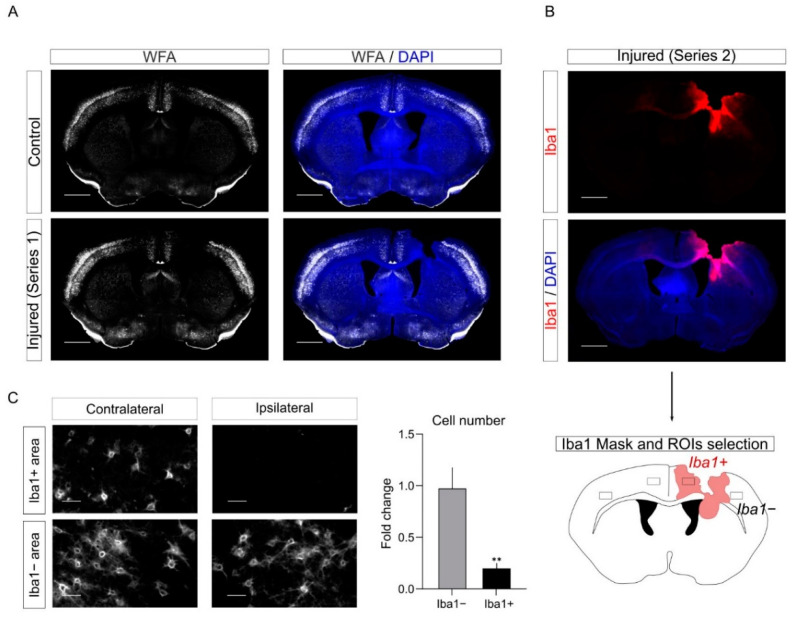
PNNs are affected in the PL upon CCD. (**A**) Representative images of WFA staining in control and injured mice. (**B**) Representative images of the Iba1 staining of a different series of the same injured animal presented in (**A**). (**C**) Representative images from the ROIs corresponding to the boxed areas in the cartoon showing the reduction of WFA labeling in the PL. The graph shows the fold change of cell number in both Iba1+ and Iba1− PLs compared to contralateral area. *n* = 12 slices. Average ± sem. ** *p* ≤ 0.01. Paired *t*-test.

## Data Availability

All reagents and additional information about the results and methodology are available on request.

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
