# Peer review of "A Novel In Vivo Model for Multiplexed Analysis of Callosal Connections upon Cortical Damage"

_ijms, 2022, doi:10.3390/ijms23158224_

Round 1

Reviewer 1 Report

In this research report, the authors developed a new pre-clinical model to study the axonal regeneration, neuronal plasticity and tissue remodeling upon controlled brain injury. They further validate their model of Controlled Cortical Damage (CCD) by providing an example of its application to assess the correlation between postlesional microglia activation and the extent of perineuronal nets surrounding cortical parvalbumin neurons.

The work combines a large panel of methodological approaches going from molecular tracing of both ipsi- and contra-lateral cortical connections and stereological analysis of histological markers to using a battery of behavioral tests as a functional read-out of CDD. Moreover, the authors developed an approach to delineate more accurately the perilesional area, which remains a relevant obstacle in the current experimental models of brain injury.

The work is rigorously conducted and controlled (e.g. by using the young mice as a control for CDD in aged mice). The paper is well written and easy to follow and represents an important advancement in the field since available models of brain injury showed relatively limited potential in testing the neuroprotective drugs. Indeed, all drugs found efficient in the current models of brain injury subsequently failed in phase II or phase III clinical trials (Petersen A et al, Exp Brain Res 2021; doi: 10.1007/s00221-021-06178-6 ).

Minor comment:

1)    Please cite the above recent review by Petersen A et al, Exp Brain Res 2021.

Author Response

Response to Reviewer #1

We are grateful to Reviewer #1 for the extremely positive and encouraging revision of our manuscript. Developing a novel experimental paradigm is always risky and time consuming. This challenge was particularly complex since different experimental models are available in the field of brain damage. Moreover, coming from a scientific background in the molecular biology of axon guidance and molecular psychiatry, it was very important for us to provide a novel and fresh viewpoint while maintaining a sound and rigorous scientific approach.

We thank the Reviewer #1 for understanding our effort and the novelty of our experimental paradigm. We particularly appreciate that the Reviewer #1 found our work “rigorously conducted and controlled” and for stating that our work “represents an important advancement in the field”.

Of course, we were happy to include the interesting Review suggested by the Reviewer by Petersen et al. among the references in the amended version of the manuscript.

Reviewer 2 Report

The authors of the manuscript A novel in vivo model for multiplexed analysis of callosal connections upon cortical damage designed an elegant and simple method to track cortico-cortical and cortical-callosal connections in mouse brain upon brain injury. 

Though I found the manuscript very interesting, a few points should be addressed in my opinion: 

1. The authors focused on Brain Injury/Controlled Cortical Damage. As the manuscript efficiently showed a method to label callosal projections, they must include a model of demyelination (LPC local injection) followed by myelin staining (fluoromylein or MBP).

2. In figure 2, the authors used the term ipsilateral referring to the site of brain injury even in absence of any traumatic procedure. The authors should perhaps change terminology in this figure

3. How different is tissue recovery in aged mice? As most studies in the field focus on recovery from brain damage, the authors should include data documenting the resolution of the cortical scar in young and aged mice 

4. The experiments of locomotor coordination are well designed. Perhaps the balance beam test can also be adopted by other scientists

5. Is the experimental design counterbalanced? Did the authors alternate side of injury?

6. I perhaps missed the staining data for GFAP and parvalbumin. Did the authors include these in the manuscript? 

Author Response

Reviewer #2 point-to-point response

NOTE TO THE EDITOR AND REVIEWER:

Some formating was lost in copying this response. Please, refer to the attached file for the correct format of this Point-to-point response.

We thank Reviewer #2 for her/his kind revision. We particularly appreciate that Reviewer #2 found our manuscript “very interesting”, “elegant and simple”. Our goal was exactly to develop a workflow that could be at once innovative, interesting, and yet relatively simple and “user friendly”. We are happy that the Reviewer valued our effort and the novelty of our approach. In addition, we are grateful to Reviewer #2 for providing constructive feedback and suggestions that further improved the quality of our work.

Here below, we provide a point-to-point response to the comments and suggestions of the Reviewer #2.

To facilitate the revision, please find in Blue the original Reviewer’s comments and in Black our response. In Green, we highlight the amendments that were added to the original manuscript.

Point-to-point response

The authors of the manuscript A novel in vivo model for multiplexed analysis of callosal connections upon cortical damage designed an elegant and simple method to track cortico-cortical and cortical-callosal connections in mouse brain upon brain injury. 

Though I found the manuscript very interesting, a few points should be addressed in my opinion: 

  1. The authors focused on Brain Injury/Controlled Cortical Damage. As the manuscript efficiently showed a method to label callosal projections, they must include a model of demyelination (LPC local injection) followed by myelin staining (fluoromylein or MBP).

This is a very interesting suggestion and we will certainly follow it up in the future. Of course, the process of demyelination and remyelination is important for the regeneration and plasticity of cortical circuits. In the context of the current manuscript, we believe that repeating the experiments with LPC injection as demyelination model would be redundant and out of the scope of the current study. Nonetheless, we modified the text in the discussion to give more importance to the topic of the changes in myelination upon brain damage.

The manuscript was amended as follows:

Discussion, Page 23, second paragraph.

In addition to axonal growth and synapse formation, the physiology of cortical circuits is further regulated by the process of axonal myelination. Deficits, in the myelination were related to impaired signal transmission and were implicated in both neurodevelopmental pathologies and brain damage [38]. The process of re-myelination during the recovery and plasticity of cortical circuits is poorly understood and the oligodendrocytes are a major cellular target for novel therapeutic approaches to brain injury.

  1. In figure 2, the authors used the term ipsilateral referring to the site of brain injury even in absence of any traumatic procedure. The authors should perhaps change terminology in this figure

We thank the Reviewer for this comment. The text of the manuscript was amended accordingly to clarify this point:

Page 6, third paragraph:

In addition, we showed that the labeling could be combined with different tracers such as biotin dextran amine (BDA). Specifically, we labeled with BDA injection in the right motor cortex the ipsilateral connections from the right forelimb (bregma 1.60; ML 1.60; DV 0.5) to the right hindlimb.

Conversely, the text in the figure legend was left unaltered since, we believe, it is correct and sufficiently clear thanks to the illustration in the Cartoons (Figure 2B’ and 2B’). The callosal connections in Green are defined here as contralateral since they connect the neurons to their respective contralateral side. Conversely, the connections in Red are defined as ipsilateral since in the schema of the Cartoon they illustrate the ipsilateral connections from the right forelimb to the right hindlimb.

  1. How different is tissue recovery in aged mice? As most studies in the field focus on recovery from brain damage, the authors should include data documenting the resolution of the cortical scar in young and aged mice 

We thank the Reviewer for this constructive comment. Of course, an extensive analysis of the differences between the responses to damage in young and old mice is out of the scope of the current manuscript. In our work, we took advantage of “young” (3-4 months old) mice to set up and characterize the protocol for the CCD primarily for ethical and practical reasons. The more in depth behavioral and histological analysis were instead performed in mice aged between 9 and 12 months. We believe it is worth noting in this regard that for studies specifically focused on aging, mice are normally considered “old” from at least 18 to 24 months.

To follow up to the feedback of Reviewer #2 we performed an additional volumetric analysis of the lesion and of the inflammatory response taking advantage of the Iba1 labeling.

We already mentioned in the text of the original version of the manuscript (page 10, third paragraph) that we did not observe relevant differences in the size of the lesion between young and aged mice. We now extended this analysis and provided a direct comparison of the volumes of the lesion in the new Supplemental Figure 1C.

Moreover, in response to Reviewer #2 we performed an additional volumetric analysis of the Iba1 positive regions in young and aged mice. Again, we did not observe major differences in the experimental groups. Nonetheless, we observed a slight tendency to have a wider Iba1 positive region in the aged mice (p=0.14). This difference failed to pass the test for statistical significance in our experimental groups.  Future studies will be required to further explore this interesting topic. Indeed, an increased inflammatory response would be consistent with the literature on the pathophysiology of aging. However, we believe that a study specifically aimed at investigating the differences in the response to damage in young and old mice would require the use of a big cohort of mice older than 18 months to mimic more closely the alterations related to aging.

The additional analysis on Iba1 related response is now reported in Supplemental Figure 1. Moreover, we added in the result and discussion specific paragraphs on this topic.

Results 2.7, Page 16, 4th paragraph:

Incidentally, we did not detect major differences between young (3-4 months) and aged (9-13 months) animals in the volume of injury and of the Iba1 positive region (Supplemental Figure 1C). Nonetheless, we observed a tendency towards increased volume of the Iba1 positive region in aged mice (Supplemental Figure 1C). In future studies, it will be interesting to evaluate whether Iba1 activation in older mice (20-24 months) is increased.

Discussion, Page 23, 3rd paragraph:

Another major challenge will be understanding the cellular and molecular mechanisms related to neuronal plasticity upon injury in the old. Multiple pathways related to survival, oxidation and inflammation are altered in neurons by ageing. However, these studies require the use of mice of at least 18-24 months of age which significantly complicates the experimental procedures and planning. Nonetheless, the development of novel preclinical models of brain injury and recovery in aged mice will be essential, considering the epidemiological impact of brain damage and related permanent disabilities in the elderly.

  1. The experiments of locomotor coordination are well designed. Perhaps the balance beam test can also be adopted by other scientists

We thank the Reviewer for this kind comment. We hope indeed that this experimental paradigm will be taken up by other groups. In particular, the motor characterization required a lot of testing and refinement before we could narrow down our experimental workflow to the assays proposed in this manuscript that are, in our experience, the most sensitive and cost effective.

  1. Is the experimental design counterbalanced? Did the authors alternate side of injury?

The control additional suggested here by the Reviewer is intriguing. Nevertheless, we would like to humbly state that, to the best of our knowledge, this control is not normally done in this kind of tests. This was also confirmed by other colleagues working in the field of brain damage. Besides, we failed to find in the literature reports suggesting a major “side preference” in the motor assay used in this study.

To respond to the Reviewer #2, we present an additional and more in depth characterization of the “side preference” in pre-injured animals. We performed routinely this assay in pre-injured animals specifically to detect anomalies in motor behavior that would not depend on the CCD. We focused on the Rotating Pole Test since it is the more adapt to reveal a “side preference”. In these assays, we could not observe any relevant difference between right or left hindlimb in the conditions tested (i.e. clockwise or counterclockwise rotation).

This dataset was inserted as an additional control in the Methodological session:

Page 24, Last paragraph

To ensure that the motor deficits were linked to cortical damage, mice performed the test a couple of days before the CCD surgery. As the model proposed conformed a unilateral lesion in the right motor cortex, we previously check whether animals showed side preference before the lesion induction. We evaluated the left and right hind limb data obtained in the rotating pole test (RPT) before the injury. The comparison between both paws was tested in rotation to the right and to the left modes (two-way ANOVA with Šídák's multiple comparisons test, adjusted p=0,809 and adjusted p=0,644 respectively). The data did not reveal statistical differences, so we discarded hind limb dominance effect in our experimental groups.

  1. I perhaps missed the staining data for GFAP and parvalbumin. Did the authors include these in the manuscript? 

We thank Reviewer #2 for this observation. Our approach allows to combine multiple molecular markers, many of them would be interesting. In this work, we focused on the PNN. Since PNN are normally enriched in PV+ interneurons, to respond to the comment of the reviewer we added in the revised version of the manuscript representative images of the PV labeling in the cortex combined with the WFA staining. This colabeling further confirms the specificity of the labeling for WFA in the cortex.

These representative images are presented as Supplemental Figure 2.